Analysis of biopsies of gastric cancer, intestinal and diffuse, and non-atrophic gastritis: an overview of loss of heterozygosity in Mexican patients

Larios-Serrato Violeta 1
Valdez-Salazar Hilda A. 2
Torres Javier 2
Camorlinga Margarita 2
Piña-Sánchez Patricia 3
Minauro Fernando 4
Ruiz-Tachiquín Martha-Eugenia mertachiquin@gmail.com 3
1 Laboratorio de Biotecnología y Bioinformática Genómica/Escuela Nacional de Ciencias Biológicas, Instituto Politécnico Nacional , Ciudad de México , Mexico
2 Unidad de Investigación Médica en Enfermedades Infecciosas y Parasitarias/Unidad Médica de Alta Especialidad-Hospital de Pediatría ‘Dr. Silvestre Frenk Freund’/Centro Médico Nacional Siglo XXI, Instituto Mexicano del Seguro Social , Ciudad de México , Mexico
3 Unidad de Investigación Médica en Enfermedades Oncológicas/Unidad Médica de Alta Especialidad-Hospital de Oncología/Centro Médico Nacional Siglo XXI, Instituto Mexicano del Seguro Social , Ciudad de México , Mexico
4 Unidad de Investigación Médica en Genética Humana/Unidad Médica de Alta Especialidad-Hospital de Pediatría ‘Dr. Silvestre Frenk Freund’/Centro Médico Nacional Siglo XXI, Instituto Mexicano del Seguro Social , Ciudad de México , Mexico
Qin Jiangjiang
Electronic publication date: 2025 Feb 25
Publication date: 2025
Volume: 13
Electronic Location ID: e18928
Received 2024 Aug 13; Accepted 2025 Jan 13
Copyright: ©2025 Larios-Serrato et al.
Copyright year: 2025
Copyright holder: Larios-Serrato et al.
License: This is an open access article distributed under the terms of the Creative Commons Attribution License, which permits unrestricted use, distribution, reproduction and adaptation in any medium and for any purpose provided that it is properly attributed. For attribution, the original author(s), title, publication source (PeerJ) and either DOI or URL of the article must be cited.
License URL: https://creativecommons.org/licenses/by/4.0/

Keywords: LOH, Diffuse gastric cancer, Intestinal gastric cancer, High density arrays, Gene markers

Funding: Fondo de Investigación en Salud-Instituto Mexicano del Seguro Social FIS/IMSS/PROT/G16/1573 FIS/IMSS/PROT/PRIO/13/027 The present study was supported by the Fondo de Investigación en Salud-Instituto Mexicano del Seguro Social (grant nos. FIS/IMSS/PROT/G16/1573 and FIS/IMSS/PROT/PRIO/13/027). The funders had no role in study design, data collection and analysis, decision to publish, or preparation of the manuscript.

==============================
This study analyzed the loss of heterozygosity (LOH) effect on gastric cancer (GC) tumor samples from 21 Mexican patients, including diffuse (DGC) and intestinal (IGC) subtypes, as well as non-atrophic gastritis (NAG, control). Whole-genome high-density arrays were performed, and LOH regions were identified among the tissue samples. The differences in affected chromosomes were established among groups, with chromosomes 6 and 8 primarily affected in DGC and chromosomes 3, 16, and 17 in IGC. Functional pathway analysis revealed involvement in cancer-associated processes, such as signal transduction, immune response, and cellular metabolism. Five LOH-genes (IRAK1, IKBKG, PAK3, TKTL1, PRPS1) shared between GC and NAG suggest an early role in carcinogenesis. Specific genes were highlighted for Hallmarks of Cancer NAG-related genes (PTPRJ and NDUFS) were linked to cell proliferation and growth; IGC genes (GNAI2, RHOA, MAPKAPK3, MST1R) to genomic instability, metastasis, and arrest of cell death; and DGC genes to energy metabolism and immune evasion. These findings emphasize the role of LOH in GC pathogenesis and underscore the need for further research to understand LOH-affected genes and their diagnostic or evolution potential in cancer management. Portions of this text were previously published as part of a preprint (https://www.medrxiv.org/content/10.1101/2024.07.29.24311063v1).

Introduction

Gastric cancer (GC) ranks fifth in the world according to its incidence and mortality (Bray et al., 2024). GC is a multifactorial disease with environmental and genetic factors affecting its occurrence and development. The GC incidence rate rises progressively with age; the average age for diagnosis is 70 years. However, a small portion of gastric carcinomas (10%) are detected at younger ages (45 or less), becoming good chances to look for GC early alterations as carcinogenesis pathways or genetic causes since those patients are less exposed to environmental influence. GC carcinogenesis is a multistage process with a progressive development, that involves gene mutations and epigenetic alterations (Machlowska et al., 2020).

GC molecular classification encloses four subtypes: chromosomal instability tumors (CIN), microsatellite instability tumors (MSI), genomically stable tumors (GS), and Epstein-Barr virus (EBV)-positive tumors (Cancer Genome Atlas Research Network, 2014). CIN is one of the most significant subtypes that involves genomic instability pathways. It is characterized by losses or gains of whole chromosomes, as well as changes in the region of chromosomes that result in aneuploidy, including allelic losses like loss of heterozygosity (LOH), gene deletions, and amplifications (Lengauer, Kinzler & Vogelstein, 1998; Ottini et al., 2006) or rearrangement (Liu & Meltzer, 2017).

Further, two main GC histotypes are recognized: intestinal and diffuse. Although most of the described genetic alterations have been observed in both types, different genetic pathways have been hypothesized. Genetic and epigenetic events, including LOH, have mostly been reported in intestinal-type gastric carcinoma (IGC) and its precursor lesions. In contrast, LOH mutation (p53) is implicated in diffuse-type gastric cancer (DGC) (Nobili et al., 2011).

LOH, a Hallmark of Cancer, has been identified as an etiological factor in CIN (Liu & Meltzer, 2017). LOH involves the loss of one of the two gene alleles in a cell, which can lead to the inactivation of tumor suppressor genes and contribute to the development and progression of cancer (Nobili et al., 2011). There are two types of LOH, copy number loss LOH (CNL-LOH) because of the diminished alleles, e.g., for tumor suppressor genes, and neutral copy number LOH (CNN-LOH), without any affecting function nor contributing to the disease development (Fitzgibbon et al., 2005). A complete or partial deletion of a chromosome leads to CNL-LOH, while CNN-LOH is mainly caused by acquired uniparental disomy (UPD) and genetic conversion and occurs without any net change in copy number (Ryland et al., 2015; Tapial et al., 2021).

Various molecular techniques have been used to investigate the LOH role on cancer, such as polymerase chain reaction (PCR) (Huo et al., 2021), microsatellite marker sites PCR (Xiao et al., 2006), multiplex ligation-dependent probe amplification (MLPA) (Milne et al., 2010), polyacrylamide gel electrophoresis (PAGE) (Zhu et al., 2017), silver stain (Chung et al., 2010), exome sequencing (Ohshima et al., 2019), Illumina (Arakawa et al., 2017) and Affymetrix microarrays (Choi, Lee & Cho, 2018). The identification of LOH events can be assessed by gene expression using RNA-Seq and reverse transcription polymerase chain reaction (RT-PCR) (Choi et al., 2018) and protein expression with immunohistochemistry (IHC) (Choi, Lee & Cho, 2018). LOH has also been correlated with CpG hypermethylation processes in GC patients (Wang et al., 2016). The most frequently reported GC-associated LOH-genes are TP53 (Battista et al., 2021), PTEN (Ohshima et al., 2019), RB1, and BRCA1 (Wu et al., 2022) on chromosomes 1, 3, 7, 8, 11, 12, 13, 18, and 22 (Nobili et al., 2011; Arakawa et al., 2017; Hosseini et al., 2007). Different scores have been proposed to establish a diagnosis or risk rate based on LOH (Wu et al., 2022). Then, our interest in determining the LOH patterns in a group of IGC, DGC, and non-atrophic gastritis (NAG) samples was to find possible guidelines for therapeutic targets and data that enriches the knowledge of cancer biology.

Materials & Methods

The Comisión Nacional de Investigación Científica (CNIC) del Instituto Mexicano del Seguro Social (Institutional Review Board, IRB) approval was obtained for the study (approval no. 2008-785-001). Clinical data and patient samples were processed following written informed consent.

Samples

Tissue samples were obtained from 21 patients (five females and 16 males) that met the criteria for diffuse gastric cancer (DGC, n = 7) and intestinal gastric cancer (IGC, n = 7) diagnoses, and subjects with non-atrophic gastritis (NAG, n = 7) as controls. Histological assessment of the biopsies was performed by two trained pathologists independently. They assigned the phenotypic diagnosis of diffuse or intestinal tumors and non-atrophic gastritis (NAG) samples. Only samples with the same diagnosis (‘identical results’) by two independent expert pathologists were included in the analysis (Table 1).

Table 1 Characteristics of gastric cancer samples analyzed in this study.

ID	Age
(years)	Sex	CT	%CC	H. pylori	TNM	Treatment	
3CG-008	72	M	Intestinal	70	Positive	I B T1 N1 M0	Naïve	
3CG-126	80	M	Intestinal	60	Negative	IIA T4 N0 M0	Naïve	
3CG-128	91	M	Intestinal	70	Negative	IIA T3 N2 M0	Naïve	
3CG-046	52	F	Intestinal	60	Negative	IV T4 N2 M0	Naïve	
3CG-099	59	M	Intestinal	50	Negative	II T3 N0 M0	Naïve	
3CG-146	71	M	Intestinal	60	Negative	IIB T3 N2 M0	Naïve	
3CG-104	69	M	Intestinal	60	Negative	III A T4 N0 M0	Naïve	
3CG-047	58	M	Diffuse	70	Negative	IV T4 N3 M0	Naïve	
3CG-173	76	M	Diffuse	70	Negative	III A T2 N3 M0	Naïve	
8CG-004	76	M	Diffuse	70	Negative	II T1 N0 M0	Naïve	
1CG-001	45	M	Diffuse	60	Negative	IV T4 N2 M1	Naïve	
3CG-035	55	M	Diffuse	60	Negative	IV T4 N2 M0	Naïve	
3CG-042	64	M	Diffuse	50	Negative	IV T4 N2 M0	Naïve	
3CG-064	38	M	Diffuse	50	Negative	IV T4 N2 M0	Naïve	
4GB-001	64	M	NAG	0	Negative	NA	NA	
4GB-031	62	M	NAG	0	Negative	NA	NA	
4GB-015	35	F	NAG	0	Negative	NA	NA	
4GB-025	39	M	NAG	0	Positive	NA	NA	
4GB-033	76	F	NAG	0	Positive	NA	NA	
4GB-036	38	F	NAG	0	Positive	NA	NA	
4GB-042	77	F	NAG	0	Negative	NA	NA	
Notes.

ID identification code

CT cancer type

CC cancer cells

M male

F female

NA not applicable

TNM, T the extent of the primary tumor

N the absence or presence and extent of regional lymph node

M the absence or presence of distant metastasis

NAG non-atrophic gastritis

To guide the investigation of relevant alterations, the present analysis focused on LOH events present in at least three patients (cut-off, ≥3 patients; ≥40% samples) to identify the most relevant GC alterations. The DNA extraction, DNA quality assessment, and high-density whole-genome processing microarray analysis were done according to Larios-Serrato et al. (2022).

LOH processing

The raw intensity files (.CEL) retrieved from the commercial platform Affymetrix® CytoScan™ microarray (Affymetrix; Thermo Fisher Scientific, Inc.) were analyzed using Chromosome Analysis Suite v4.3.0.71 (ChAS, https://www.thermofisher.com/). The construction of the GRCh38 genome (December 2013) was used as a reference model and CytoScanHD_Array.na36.annot.db file for annotation. Data processing was based on the segmentation algorithm, where the Log2 ratio for each marker was calculated relative to the reference signal profile. To calculate the LOH, the data were normalized to baseline reference intensities using the ChAS reference model, including 284 HapMap samples and 96 healthy individuals. The Hidden Markov Model (HMM) was used to determine the LOH segment calls. The customized conditions were filtered to determine LOH, 3 Mb, and 50 single nucleotide polymorphisms (SNPs). The median absolute pairwise difference (MAPD) and the single nucleotide polymorphism quality control (SNPQC) score were used as the quality control parameters. Only samples with MAPD < 0.25 and SNPQC > 15 values were included in the further analysis.

Bioinformatics analysis

To generate a list of genes and frequencies for altered regions, Practical Extraction and Report Language (Perl) scripts (Wall, Christiansen & Orwant, 2000) were developed to load the LOH segment data files generated by ChAS v4.3.0.71 for each sample, including chromosomes, cytogenetic bands, Online Mendelian Inheritance in Man information, and haploinsufficiency predictions version 3 information from the DatabasE of Genomic Variation and Phenotype in Humans, using Ensembl Resources (DECIPHER v11.25). Custom scripts were developed in Perl v5.32 to obtain the frequency of LOH genes and cytobands, and the length of events.

The genes altered in three patients (cut-off, ≥ 3) with DGC, IGC, and NAG were included for analysis and visualization. The genes were compared by generating Venn diagrams with the Jvenn server (Bardou et al., 2014). Cancer Hallmarks enrichment analysis (p.adjust < 0.05) was performed with a collection of 6,763 genes (Menyhart, Kothalawala & Győrffy, 2024), available on the server http://www.cancerhallmarks.com; this database establishes the relationship of genes and Hallmarks through the collection of different publications. The Cancer Hallmarks server uses a hypergeometric test for enrichment using the greasy Python package. The results were reviewed using the Catalog of Somatic Mutations in Cancer database (COSMIC v100) (Sondka et al., 2024) and the Hallmarks of Cancer database (HOCdb database). Reactome v88 performed a metabolic pathway enrichment analysis (Fabregat et al., 2018), considering those results significant with values less than 0.05 in the false discovery rate (FDR).

Finally, an interaction network was generated based on metabolic pathways, genetics, and physical and functional associations to establish the cancer Hallmarks associated with the profile of LOH-genes IGC, NAG, and “core” IGC-DGC-NAG. Furthermore, these were determined using the STRING v12.0 prediction server (Szklarczyk et al., 2021) and Cytoscape v.3.10.0 (Shannon et al., 2003), including manual annotation of their corresponding cancer Hallmarks (adhesion, angiogenesis, inflammation, migration, metastasis, morphogenesis, proliferation, and survival).

Results

Sample characteristics

This study included tissue samples from 21 Mexican patients without treatment (naïve). Patient samples included seven DGC cases, seven for IGC, and seven more corresponding to NAG (as controls). The .CEL files and their raw intensity values obtained from the microarrays were deposited in the Center for Biotechnology Information (NCBI), with the accession key GSE117093 and BioProjet PRJNA481039.

Table 1 shows the general characteristics of the 21 patient samples, age (mean ± SD, 59.61 ± 15.94 years), sex (female 23.8% and male 76.2%), and the percentage of neoplastic cells for tumor tissues ranging between 50 and 70%. One IGC patient and three with NAG were positive for H. pylori. Data from the tumor size, number of nodes, and metastasis (TNM) classification system are presented.

Genomic detection of LOH

The LOH of the patients was estimated using the analysis described before, a meticulous process based on regions where the preponderance of SNPs does not display heterozygosity. Table 2 shows a summary of the chromosomes with the highest involvement frequency concerning the number of events (coincidences) at the LOH-regions, but not strictly perfect in the chromosomal coordinates.

Table 2 Top of altered cytobands in diffuse gastric cancer, intestinal gastric cancer, and non-atrophic gastritis.

Type	Cytoband	Length (Mbp-cl)	Patient
numbers	
DGC	Xp22.33	338.81	7	
Xq11.1	651.48	7	
16p11.2	26.93	6	
IGC	Xq11.1	496.05	7	
16p11.2	22.84	6	
Xp22.33	333.25	6	
3p21.31	19.31	5	
17q22	10.61	3	
NAG	Xq11.1	385.68	7	
16p11.2	30.43	7	
Xp22.33	222.15	7	
Notes.

DGC Diffuse gastric cancer

IGC Intestinal gastric cancer

NAG Non-atrophic gastritis

Mbp-cl Megabase pairs cumulative length

Our data, which include the megabase pairs cumulative length (Mbp-cl) of our tissue samples, were also reviewed (Table S1). The LOH-gene frequency data, chromosomes, and cytobands are presented. Table S2 displays the accumulated LOH-length (Mb) values per chromosome to determine if more extended losses indicate more damage.

In DGC patients, the affected chromosomes with Mbp-cl and the specified number of LOH-events were 6, 8, 16, and X; at IGC, they were chromosomes 3, 16, and 17. Chromosomes 6 and 8 are associated with DGC, while 3 and 17 are associated with IGC (Table 2 and Table S3). Following, we found that there are 3,361 LOH-genes in DGC (Table S1); chromosomes Xq11.1/Xp22.23 in 7/7 male patients and chromosome 16p11.2 in 6/7 male patients (Table 2 and Table S3) were the most altered.

Figure 1 Profile of LOH-genes in gastric cancer from ≥ 3 patients.

(A) The Venn diagram presents frequencies of specific and shared genes in diffuse gastric cancer (DGC), intestinal gastric cancer (IGC), or non-atrophic gastritis (NAG) in LOH. Each subset of the Venn diagram in parentheses indicates the Cancer Hallmark genes (B, C, and D). In the radial graphs, each bar represents a Hallmark and the height of the p.adjust value of the category enrichment; the grays have no significance. The red dotted line represents the significance cut-off value (p.adjust < 0.05).

Table 3 Metabolic pathways and genes-Hallmarks of cancer.

	General description	Pathway	p -value	Genes	
NAG	Signal by receptor tyrosine kinases	Signaling by MET
Signaling by VEGF	2.2E−02
2.6E−02	PTPRJ	
Signal transduction	RHOG GTPase cycle	7.2 E−03	NDUFS3	
IGC	Signal transduction	Signaling by GPCR
Signaling by WNT
Signaling by MST1	9.3E−03
7.9E−04
7.9E−04	GNAI2
CLTC, WDR6
MST1, MST1R	
Developmental biology	Semaphorin interaction (Sema 4D)	1.4E−03	PLXNB1, RHOA	
Extracelullar matrix organization	Laminin interactions
ECM proteoglycans	2.9E−04
1.2E−02	LAMB2, COL7A1
LAMB2, DAG1	
Cell cycle	Activation of ATR in response to replication stress	2.2E−02	ATRIP, CDC25A	
Immune system	Regulation by TREX1	3.2E−02	TREX1	
Proteins and carbohydrates Metabolism	Post-translational protein phosphorylation
Hyaluronan uptake and degradation	7.8E−03
7.6E−03	LAMB2, SHISA5, HYAL1, HYAL2	
DGC-IGC-NAG	Immune system	Toll-like receptor cascades	9.0E−03	IRAK1, IKBKG, TAB3	
Signal transduction	Class A71 rhodopsin like receptors
RHO GTPase cycle	6.4E−04
1.3E−02	OPN1MW, OPN1LW2x
OCRL, OPHN1, STARD8, WAS, PAK3, FGD1, ARHGEF6, ARHGAP4	
Extracellular matrix organization	Collagen Formation	2.2E−02	COL4A6, COL4A5	
Metabolism	Aerobic respiration and electron transport
5-Phosphoribose 1-diphosphate biosynthesis
Diseases associated with glycosaminoglycan	3.0E−02
1.7E−02
1.6E−02	PDHA1, PDK3
PRPS2, PRPS1
GPC3, GPC4	
Cellular response to stimuli	KEAP1-NFE2L2	5.1E−03	TKTL1, G6PD	
Cell cycle	Cohesin loading onto Chromatin (M phase)	1.0E−02	STAG2, SMC1A	
Others
	Developmental biology
RSK activation
WNT cancer	29E-04
1.3E−03
1.5E−03	MECP2, TBL1X
RPS6KA3, RPS6KA6,
PORCN, AMER1	

Figure 2 Relevant gene networks, and the Hallmarks of Cancer and metabolic pathways.

Ten Hallmarks are shown in the legend. The IGC-DGC-NAG network is indicated in purple, NAG in pink, and IGC in blue.

In IGC, 2,490 LOH-genes were determined (Table S1) with chromosomes Xq11.1 for 7/7 patients, Xp22.33 for 6/7 patients, 16p11.2 for 6/7 patients, 3p21.31 for 5/7 patients and 17q22 for 3/7 patients (Tables 2 and S3) as the most altered.

Finally, in NAG, 4,748 LOH genes were determined (Table S1). Chromosomes Xq11.1 for 7/7 patients, 16p11.2 for 7/7 patients, and Xp22.33 for 4/7 patients (Tables 2 and S3) were the most altered.

Interestingly, LOH lengths do not seem relevant to carcinogenesis, and 5–10 Kbp LOH lengths were more common and frequent in DGC, IGC, and NAG (Table S4).

We analyzed alterations occurring in at least three patients to identify the most relevant LOH in GC and NAG (cut-off ≥ 3). We found a similar pattern for total LOH, with events in DGC (1157), IGC (1361), and NAG (1184). In addition, DGC had the highest number of genes affected in all samples, 7/7 (1132), followed by IGC (261) and NAG (34) (Table S1).

Gastric cancer genes associated with LOH

A Venn diagram was constructed to examine the LOH-GC-relevant genes of at least three patients (cut-off ≥ 3) of the DGC, IGC, and NAG. We determined 1,153 shared LOH genes between DGC-IGC-NAG. IGC had 207 unique affected genes, while NAG only showed 28 (Fig. 1A and Table S5). From each subset (Fig. 1A), those genes with matches according to the Cancer Hallmarks Genes database, a comprehensive resource that includes 6,763 genes, are shown 241 LOH-genes were found in DGC-IGC-NAG; IGC had 55 affected unique genes and 13 genes in NAG. Figures 1B–1D represent the enrichment of the Hallmarks of Cancer. The 241 common genes DGC-IGC-NAG present more Hallmarks than NAG, which showed fewer.

Functional pathway analysis

LOH-genes-Hallmarks were used to identify metabolic pathways in each subset according to Reactome and Homo sapiens as a model organism (Table 3); those significant metabolic pathways (p-value < 0.05) were selected as the most affected.

Correlation genes network

Cancer LOH-genes-Hallmarks associated with metabolic pathways were used to construct interaction networks (STRING, Fig. 2). The connecting lines indicate associations by metabolic pathways, expression, localization, inferred interaction, genetic interactions, data mining, and neighborhood. Likewise, each node can be related to flags (events) reporting the Hallmarks of Cancer. In this way, the network formed among IGC-DGC-NAG had 31 nodes; the genes with the highest number of Hallmarks were PAK3, IRAK1, and IKBKG, and the genes with the highest number of connections were OPHN1, WAS, TKTL and PRPS1. In the ICG network, there were 29 nodes. The genes associated with cancer Hallmarks were GNAI2, RHOA, MAPKAPK3, HYAL1, and CISH, while the most connected were RHOA, MST1R, and ATRIP. Finally, the network corresponding to NAG had five nodes, PTPRJ had the most significant number of Hallmarks, and the most connected was NUP160 (Table 4).

Discussion

Design study

Our study’s choice of seven patients per group is based on rigorous selection criteria. Although our repository has approximately 600 GC samples, only those with at least 50% cancer cells (see Table 1) were included to reduce the “noise” from non-cancerous cells, which is essential for accurate results. In molecular biology and genetic studies, it is expected to face logistical and financial constraints that limit the collection of large sample sizes. Additionally, in highly controlled experiments, a smaller sample size can be sufficient to detect significant effects. Despite a limited “n”, wide confidence intervals help interpret the precision of our results. Furthermore, previous studies in the literature have used similar sample sizes and support our methodology. Finally, as an exploratory study, this research provides a foundation for larger-scale studies to generate valuable hypotheses for future experiments (Larios-Serrato et al., 2022).

Table 4 Top genes linked to the highest number of the Hallmarks of cancer and genes detected in the stomach.

Tissue	Gene, ID	Interaction number	Cytoband	Function	
NAG	PTPRJ, 5795	3	11p11.2	The protein encoded by this gene is a protein tyrosine phosphatase T(PP) family member. PTPs are signaling molecules that regulate a variety of cellular processes, including cell growth, differentiation, the mitotic cycle, and oncogenic transformation.	
NDUFS, 4722	2	11p11.2	This gene encodes one of the iron-sulfur protein (IP) components of mitochondrial NADH: ubiquinone oxidoreductase (complex I). Mutations in this gene are associated with Leigh syndrome, which results from mitochondrial complex I deficiency.	
NAG-IGC-DGC	PAK3, 5063	5	Xq23	The protein is a serine-threonine kinase and forms an activated complex with GTP-bound RAS-like (P21), CDC2 and RAC1. This protein may be necessary for dendritic development and the rapid cytoskeletal reorganization in dendritic spines associated with synaptic plasticity.	
IRAK1, 3654	5	Xq28	This gene encodes the interleukin-1 receptor-associated kinase 1. This gene is partially responsible for IL1-induced upregulation of the transcription factor NF-kappaB.	
IKBKG, 8517	6	Xq28	This gene encodes the regulatory subunit of the inhibitor of kappaB kinase (IKK) complex, which activates NF-kappaB and activates genes involved in inflammation, immunity, cell survival, and other pathways.	
TKTL1, 8277	6	Xq28	The protein encoded by this gene is a transketolase that acts as a homodimer and catalyzes the conversion of sedoheptulose 7-phosphate and D-glyceraldehyde 3-phosphate to D-ribose 5-phosphate and D-xylulose 5-phosphate. This reaction links the pentose phosphate pathway with the glycolytic pathway.	
PRPS1, 5631	5	Xq22.3	This gene encodes an enzyme that catalyzes the phosphoribosylation of ribose 5-phosphate to 5-phosphoribosyl-1-pyrophosphate. This process is necessary for purine metabolism and nucleotide biosynthesis. Defects in this gene cause phosphoribosylpyrophosphate synthetase superactivity.	
IGC	GNAI2, 2771	7	3p21.31	The protein encoded is an alpha subunit of guanine nucleotide-binding proteins (G proteins). The protein is involved in the hormonal regulation of adenylate cyclase.	
RHOA, 387	7	3p21.31	This gene encodes a member of the Rho family of small GTPases, which function as molecular switches in signal transduction cascades. Rho proteins promote reorganization of the actin cytoskeleton and regulate cell shape, attachment, and motility. Overexpression of this gene is associated with tumor cell proliferation and metastasis.	
MAPKAPK3, 7867	5	3p21.2	This gene encodes a member of the Ser/Thr protein kinase family. This kinase functions as a mitogen-activated protein kinase MAP kinase. MAP kinases, also known as extracellular signal-regulated kinases (ERKs), act as an integration point for multiple biochemical signals. This kinase was shown to be activated by growth inducers and stress stimulation of cells.	
MST1R, 4486	9	3p21.31	This gene encodes a cell surface receptor for macrophage-stimulating protein (MSP) with tyrosine kinase activity. This protein is expressed on the ciliated epithelia of the lung’s mucociliary transport apparatus and, together with MSP, is thought to be involved in host defense.	

Chronic gastritis is associated with inflammatory and cellular changes that can lead to GC. Notably, H. pylori infection can cause atrophic chronic gastritis, a precursor to gastric cancer (Kumar et al., 2020). Comparing gastritis with cancer samples can help explore the inflammation-to-cancer transition and the molecular changes involved in the progression of premalignant lesions to malignancies (Sui et al., 2020). Correa’s model describes gastric epithelium progression from chronic gastritis to intestinal metaplasia, dysplasia, and eventually adenocarcinoma (Correa & Piazuelo, 2012). It is relevant to use gastritis as a comparison group, since it represents an early phase within this cancer evolution model. Given the difficulty in collecting healthy gastric tissue due to ethical concerns with biopsies in asymptomatic individuals, gastritis is a clinically relevant control. Functionally, comparing inflamed (gastritis) tissues with cancerous ones may highlight genes or pathways that distinguish non-neoplastic inflammation from tumor-progressing inflammation, providing critical insights into tumorigenic transformation mechanisms.

LOH, chromosomes, and cytogenetics

GC has a high mortality rate due to its characteristics, such as late detection and silent progression; therefore, research into tumor biology is required to find early markers and carry out an opportune intervention. LOH is among the aspects that could contribute to a timely diagnosis according to prognosis (Huo et al., 2021; Battista et al., 2021). It has been proposed that LOH could function as independent prognostic markers (Koo et al., 2004), and those could even function as alternative targets for treatment (Hwang et al., 2021).

LOH is involved in different cancer types, showing its importance as a “predisposing” factor. According to other studies, specific LOH genes are more relevant than the length of affection. LOH-genes-NAG, TP53, PTEN, RUNX3 (Bellini et al., 2012; Li et al., 2005; Carvalho et al., 2005) could have a carcinogenic potential as signaling early events; of these, TP53 (Bellini et al., 2012) and PTEN (Li et al., 2005) have polymorphisms or copy number variations due to LOH-events (Battista et al., 2021; Poremba et al., 1995; Oki et al., 2005).

Here, 11 LOH-genes were determined, besides the small sample which were selected according to their apparition frequency in the analyzed samples, their participation in metabolic pathways (p-value < 0.05), their established interactions (networks), and their enrichment in Cancer Hallmark’s genes database (p.adjust < 0.05). Thus, PTPRJ and NUP160 were determined into NAG samples, RHOA, GNAI2, and MAPKAPK3 for ICG, and no unique or relevant genes were identified for DCG. NAG LOH-genes that are relevant to carcinogenesis participate in proliferation and growth, while those for IGC are on genomic instability, tissue invasion, metastasis, and the arrest of cell death; and DGC genes are for energy metabolism, destruction of immune evasion, and replicative immortality. Other genes were shared between IGC and NAG-IGC-DGC, whose p-values are close and could be considered similar LOH events, since they are involved in sustained angiogenesis. On the other hand, IGC genes also promote inflammation, and although the p-values are not significant, there was a difference in the NAG-IGC-DGC group. Then, those molecular, cellular, and metabolic LOH alterations should be monitored in GC patients. These findings must be validated to develop tests with molecular profiles for diagnosis, prognosis, and response to treatment, as well as, most importantly, screening tests.

When analyzing the metabolic pathways associated with LOH genes, the common ones were signal transduction, immune system, cell cycle, and extracellular matrix organization, which would be involved in early GC stages. Developmental biology is added to IGC samples because of the semaphorins interactions (Table 3, genes PLXNB1 and RHOA), which have been found to function as tumor suppressors and inhibit tumor progression by various mechanisms; however, they also can function as inducers and promoters of tumor progression (Neufeld et al., 2016). Another metabolic pathway found here (IGC) was protein and carbohydrate metabolism, where processes such as post-translational modifications (phosphorylations) and hyaluronan metabolism (degradation proteins Hyal 1–3) are affected. Hyaluron plays a fundamental role in tissue architecture and the regulation of cellular function, which could be related to proliferation and migration, since hyaluronan accumulation at the extracellular matrix and its derived fragments due to the altered expression of hyaluronidases enhance cancer progression, remodeling the tumor microenvironment (Kobayashi, Chanmee & Itano, 2020), just like in GC (Vizoso et al., 2004).

Marker genes

The results described above determined five shared genes among the three sample groups analyzed (IGC-DGC-NAG): IRAK1, IKBKG, PAK3, TKTL1, and PRPS1. These genes are associated with various typical functions of cancer development and progression. IRAK1 is involved in the interleukin 1 receptor, then regulates inflammation genes in immune cells. It is one of the most emerging Hallmarks of Cancer (because of avoiding immune destruction or tumor inflammation), affecting the disease development due to its possible oncogenic and immunological functions (Liu et al., 2022). IKBKG is linked to the NF-kB pathway, which is crucial for cancer since its involvement in cell survival and proliferation (Yin, Wang & Lu, 2020; Gong et al., 2020) is another of the principal Cancer of Hallmarks. PAK3 can regulate cell growth and migration, potentially contributing to metastasis; it can also regulate Circ 0000190, a circular RNA that inhibits CG through the caspase-3, p27, and cyclin D axis (Wang et al., 2020).

On the other hand, TKTL1 is associated with metabolism reprogramming, since it is involved in glucose metabolism in GC cells (Kämmerer et al., 2015). Moreover, it regulates other events such as proliferation, metastasis, epithelial-mesenchymal transition, resistance to chemoradiotherapy, and survival (Ahopelto et al., 2020). Finally, the relapse-specific mutations in phosphoribosyl pyrophosphate synthetase 1 gene (PRPS1), a rate-limiting purine biosynthesis enzyme that confers significant drug resistance to combination chemotherapy in acute lymphoblastic leukemia (Wang et al., 2018), suggest that these genes shared could indicate a core of early genes, given that they are also found in NAG, a non-cancerous lesion.

The NDUFS3 gene, which encodes a subunit of complex I of OXPHOS, was also identified among the shared genes. This gene also plays a significant role in metabolic reprogramming, an early event in carcinogenesis that occurs in the first phases of disease development. Metabolic reprogramming is a crucial aspect of the Warburg effect. The common feature of this altered metabolism is increased glucose uptake and fermentation of glucose to lactate, which is a common feature in cancer. Also, the H. pylori infection, like some of the NAG samples in this study, shows increased complex I, further highlighting the role of NDUFS3 (Feichtinger et al., 2017) and metabolic reprogramming as an early carcinogenesis event. Furthermore, in the MKN28 and MKN45 cell lines derived from moderately differentiated tubular adenocarcinoma and undifferentiated adenocarcinoma of medullary type, a decrease in this complex I of the mitochondrial respiratory chain is also observed (Puurand et al., 2012).

According to the NAG interaction network, PTPRJ and NUP160 genes could regulate cellular and molecular processes contributing to inflammation, immune response, energy and metabolism, and proliferation. PTPRJ encodes a protein from the tyrosine phosphatase family and dephosphorylates CTNND1, related to cell proliferation, adhesion, and migration (Du & Grandis, 2015; Holsinger et al., 2002), as with inflammation and regulation of the immune response. NUP160 encodes for a nuclear porin complex protein that regulates the transport of macromolecules between the nucleus and the cytoplasm (Xu & Powers, 2009); its deregulation can modify gene expression and cell signaling.

IGC network (Fig. 2) shows four multiple interaction genes in cancer signaling and progression process; RHOA encodes for a protein of the Rho-GTPase family and is associated with signaling (Nam, Kim & Lee, 2019), Ras homologous A (RHOA) as a significant signaling hub in GC regulates several cellular processes, including the cytoskeletal structure and cell adhesion. Recently, it was shown that RHOA can be targeted by small molecule inhibitors in cancer, implicating it as a potential druggable target (Kaibuchi, Kuroda & Amano, 1999). Therefore, it is linked to cancer progression. GNAI2 encodes the alpha subunit of the Guanine nucleotide-binding proteins (G proteins), and in stomach adenocarcinoma it is considered a prognostic marker (Li et al., 2024). Also, it is essential for the transduction of extracellular signals and is related to cellular processes such as growth, migration, regulation of proliferation, differentiation, and the response to external stimuli. MAPKAPK3 has been proposed as a therapeutic target (Cazes et al., 2022) that regulates cell stress response, proliferation, and survival; it has also been reported as an element of diagnosis, prognosis, and prediction (Niloofa, De Zoysa & Seneviratne, 2021).

Moreover, the MST1R (or RON) gene encodes macrophage-stimulating receptor-1 or Ron family tyrosine kinase receptors, which can act in the HIF-1α and β-catenin pathways. This receptor is involved in oncogenesis by regulating cell migration, adhesion, and survival; it plays an essential role in the inflammatory response and metastasis. In this way, RHOA and GNAI2 are related to the signaling and regulation of critical cellular processes in cancer. In contrast, the MST1R/RON gene is involved in the tumor progression of various types of cancer, and GC mutations have been reported (Purwar et al., 2023).

LOH and clinic

In certain tumor types, the LOH impacts over 20% of the genome, diminishing allelic diversity across numerous genes (Zhang & Sjöblom, 2021). Clinically, mutations associated with LOH on specific chromosomes can serve as biomarkers for assessing cancer risk and prognosis, may guide the development of new therapies, and offer insights into its evolution (Cavenee et al., 1983). LOH analysis is particularly relevant in oncology and is a biomarker in breast (Kaur et al., 2018), ovarian (Ryland et al., 2015), and leukemia cancers (Fitzgibbon et al., 2005). It is also linked to hereditary cancer syndromes, making it valuable in risk assessment programs for individuals with a family history of cancer (Lozac’hmeur et al., 2024).

Challenges in LOH

The study of LOH in cancer presents significant methodological and biological challenges. Variability in detection platforms, such as SNP arrays and genomic sequencing, leads to inconsistent sensitivity levels, increasing the risk of false positives or negatives. Additionally, distinguishing LOH from complex structural variants, like duplications or point mutations, complicates precise identification. Cellular context further challenges LOH analysis; it does not permanently fully inactivate tumor suppressor genes, mainly if the remaining gene copy retains partial function. Genes with haploinsufficiency add complexity, as LOH may not yield significant functional consequences. Contamination from non-cancerous cells in solid tumor samples also affects LOH detection accuracy. In contrast, genetic diversity in polyclonal tumors can skew LOH frequency estimates, complicating the interpretation of its relevance to tumor progression. Given the scarcity of LOH-focused studies, further research is essential to identify LOH-affected genes and understand their roles in cancer, potentially aiding diagnostic and therapeutic advances (Zhang & Sjöblom, 2021).

Conclusions

The frequency of LOH in specific regions of chromosomes suggests that these loci contain critical regions for tumor suppression or progression. The loss of regions where the sequences of master molecules controlling gene expression (coding and non-coding) are located plays a role in controlling the cell cycle and apoptosis. In contrast, the loss of genes can affect cell adhesion and signaling. These findings support the hypothesis that LOH contributes to genetic instability and GC progression. LOH is an essential mechanism in the oncogenesis of GC that should continue to be investigated. Identifying LOH genes could provide new opportunities for developing targeted therapeutic strategies and improving the prognosis of patients with GC. Future studies should focus on the functional characterization of these genes and the development of methods to prevent or reverse LOH in precancerous and cancerous cells.

Supplemental Information

Supplemental Information 1 Number of alterations and acumulated length in Mb, total data show before and after applying an event cut-off ≥3 patients

Supplemental Information 2 LOH genes in DGC

To identify relevant genes (cut-off, ≥3) cells are filled with gray.

Supplemental Information 3 Frequency of LOH in cytobands

The information was organized by event numbers.

Supplemental Information 4 LOH size events

Supplemental Information 5 List of genes (cut-off ≥ 3 patients) affected by LOH in Venn diagram

Supplemental Information 6 MAIME checklist

Supplemental Information 7 STROBE checklist

The authors would like to thank Mr. Brian-Alexander Cruz-Ramírez and Ms. Alejandra García-Bejarano, Unidad de Investigación Médica en Enfermedades Oncológicas, Unidad Médica de Alta Especialidad-Hospital de Oncología, Centro Médico Nacional Siglo XXI, Instituto Mexicano del Seguro Social for their technical assistance.

Additional Information and Declarations

Competing Interests

Author Contributions

Human Ethics

Data Availability

The authors declare there are no competing interests.

Violeta Larios-Serrato conceived and designed the experiments, analyzed the data, prepared figures and/or tables, authored or reviewed drafts of the article, and approved the final draft.

Hilda A. Valdez-Salazar performed the experiments, analyzed the data, prepared figures and/or tables, authored or reviewed drafts of the article, and approved the final draft.

Javier Torres performed the experiments, authored or reviewed drafts of the article, and approved the final draft.

Margarita Camorlinga performed the experiments, authored or reviewed drafts of the article, and approved the final draft.

Patricia Piña-Sánchez performed the experiments, authored or reviewed drafts of the article, and approved the final draft.

Fernando Minauro analyzed the data, prepared figures and/or tables, authored or reviewed drafts of the article, and approved the final draft.

Martha-Eugenia Ruiz-Tachiquín conceived and designed the experiments, analyzed the data, prepared figures and/or tables, authored or reviewed drafts of the article, and approved the final draft.

The following information was supplied relating to ethical approvals (i.e., approving body and any reference numbers):

Comisión Nacional de Investigación Científica (CNIC) del Instituto Mexicano del Seguro Social. Approval number: 2008-785-001

The following information was supplied regarding data availability:

The data is available at GSE117093 and BioProjet PRJNA481039.

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
