# Peer review of "Analysis of biopsies of gastric cancer, intestinal and diffuse, and non-atrophic gastritis: an overview of loss of heterozygosity in Mexican patients"

_PeerJ, doi:10.7717/peerj.18928_

## Round 0.1 · original submission · Major Revisions

Please carefully read the comments and suggestions from the reviewers and address them accordingly.

Reviewer 1 ·

Basic reporting

no comment

Experimental design

no comment'

Validity of the findings

no comment

Additional comments

The mauscript was written well. Besides, considerring the workload in bioinformatics study, the manuscript is recommended to be accepted by the journal PeerJ. However, concerns should be considered as follows:
1) the manuscript should add one paragraph discussing the limitations in the Discussion section.
2) Some errors were found when reading line 271-274 belonging to the Discussion section.
3) The sample number is too small, this issue should be discussed.
4) The formation all the references should be checked carefully according to the request of the journal PeerJ.
5) For the legend of Tables, "Table V" should be revised as "Table 5".

Reviewer 2 ·

Basic reporting

The manuscript is written in clear and professional English, and technical terms are explained adequately. However, some minor improvements in phrasing could enhance readability, especially in the introduction. For example, the explanation of molecular classification of GC could be more concise.

Experimental design

The study presents original research on a Mexican cohort of gastric cancer patients. The analysis of LOH in both gastric cancer subtypes (intestinal and diffuse) as well as non-atrophic gastritis (NAG) is novel and fills a gap in understanding gastric carcinogenesis in this population.

The methodology is rigorous, especially the detailed bioinformatics analysis used to identify LOH genes. However, more detailed explanation of how LOH was correlated with cancer hallmarks would improve replicability. Additional detail on the patient sample characteristics, beyond just age and sex, would provide further context for the study.

Validity of the findings

The study’s findings are supported by adequate statistical analysis and visualization. However, while the identification of shared genes between different patient groups is compelling, the interpretation of the relevance of these genes could be expanded. For instance, deeper exploration of the implications of certain LOH events for clinical practice (e.g., early diagnostic markers or therapeutic targets) would strengthen the discussion.

Additional comments

The authors should consider expanding the discussion of how their findings compare to other populations and geographic regions, especially since gastric cancer shows variation globally.

Reviewer 3 ·

Basic reporting

This article collected clinical samples of non atrophic gastritis, diffuse gastric cancer, and intestinal type gastric cancer, identified LOH, and analyzed cancer genes and signaling pathways to determine the genes with the greatest changes.

(1) English proficiency: The English of the article still needs to be polished, such as changing phrases like "become good chances" and "environmental facts" to more professional ones.

(2) Reference accuracy: The references are generally accurate, but the methodology section lacks references such as "LOH processing".

(3) Content: Screening for LOH genes in gastric cancer is a popular topic, but this study lacks validation experiments, which limits its significance. Meanwhile, the experimental design flaws and small sample size make the article appear lacking in persuasiveness.

Experimental design

(1) Ethics: This study obtained ethical batch numbers, informed consent forms from patients, and ethical integrity.

(2) Introduction: Firstly, the epidemiology of gastric cancer was described, and the correlation between the occurrence of gastric cancer and LOH was elucidated. Finally, the purpose of this article was introduced: to study three different LOH patterns in gastric cancer samples and identify targets. Explain clearly, use a large amount of literature support, and finally reasonably introduce the purpose of this article.

(3) method:
1) Sample screening: In this study, were the 7 cases in each group randomly selected? This article does not specify the exclusion and inclusion criteria, nor how to screen out each group of samples. I remain skeptical about the scientific and rigorous nature of this article.

2) Sample size: The sample size of this article is relatively small, and there are generally large differences in clinical samples. For omics analysis, there are at least 10 or more samples in each group, so I believe that the sample size of this article is too small and the persuasiveness of the results is lacking.

3) Group design: The author needs to introduce in the article why the NAG group is used as the control group, please explain and provide references.

4) Patient characteristics: The sample size of this article is relatively small, and the author provided a brief description of the basic characteristics. In order to reduce individual differences between clinical samples, I believe the author should conduct statistical analysis on patient characteristics and exclude the influence of other variables on the experiment.

Validity of the findings

(1) Innovation: This article only discovered the LOH gene through omics methods, and the significance of such results is very limited because there is a lack of validation experiments, which often becomes the key to readers' lack of interest.

(2) Reliable data: All data is provided and analyzed correctly.

(3) Clear conclusion: The conclusion is not clear enough and cannot include a discussion or outlook for future research.

Additional comments

(1) Image resolution: The image resolution is not high, please provide clear images.

(2) Abstract: The abstract needs to include all experimental results, please remove non experimental results and future prospects: "Our findings must be validated to develop molecular tests for 75 diagnosis, prognosis, treatment response, and, most importantly, screening tests.“

·

Basic reporting

no comment

Experimental design

1. Please specify and clarify your position about non-atrophic gastritis as controls (but not the additional comparison group) because its material and findings are also abnormal.
2. Please specify gastritis as more as possible

Validity of the findings

1. Lack of data on the degree of differentiation (grade) of intestinal-type gastric cancer. Whether this parameter could influence (be related to) the results obtained. Please indicate your opinion in the discussion.
2. Strongly different status between samples in terms of ΤΝΜ, especially reflecting the metastasis status (N, M). This probably influenced the results. Please indicate your opinion in the discussion.
3. Please outline more explicitly the position regarding the interpretation of the findings in terms of Warburg's effect. Add a little more details.

Additional comments

1. The actual small number of samples represents a significant limitation in the interpretation of the findings. Please indicate this point in the discussion.
2. Table V is probably cumbersome and the text should be simplified. Perhaps additional columns should be added.

---

## Round 0.2 · Minor Revisions

Please further address the comments from reviewers.

Reviewer 4 is saying that there are many causes of gastritis but the authors have only distinguished between cancer and non-cancer. They are asking for more info on the non-cancer cases.

Reviewer 1 ·

Basic reporting

The authors have already revised the last manuscript. And the revised manuscript has been read better than the last edition.

Experimental design

The authors have already answer the question which I have answered in the last comments. The small sample number may not influence the conclusion of the work.

Validity of the findings

The findings of the manuscript may improve the understanding of loss of the heterozygosity in Mexican patients with gastric cancer, intestinal and diûuse, or non-atrophic gastritis.

·

Basic reporting

no comment

Experimental design

Within the framework of the earlier remark about clarification of gastritis, it was first of all implied that it was necessary to clarify what kind of non-atrophic gastritis occurred in the presented patients. First of all, from the etiological point of view.

Validity of the findings

no comment

Additional comments

no comment

---

## Round 0.3 · accepted · Accept

The authors have addressed all of the reviewers' comments.